# Investigating Local Patterns of Mumps Virus Circulation, Using a Combination of Molecular Tools

**DOI:** 10.3390/v15122420

**Published:** 2023-12-13

**Authors:** Ana M. Gavilán, Paula Perán-Ramos, Juan Carlos Sanz, Luis García-Comas, Marta Pérez-Abeledo, Ana M. Castellanos, José M. Berciano, Noemí López-Perea, Josefa Masa-Calles, Juan E. Echevarría, Aurora Fernández-García

**Affiliations:** 1Centro Nacional de Microbiología, Instituto de Salud Carlos III, 28029 Madrid, Spain; agavilan@externos.isciii.es (A.M.G.); jmberciano@isciii.es (J.M.B.); 2CIBER de Epidemiología y Salud Pública (CIBERESP), Instituto de Salud Carlos III, 28029 Madrid, Spain; juan.sanz@salud.madrid.org (J.C.S.); nlopezp@isciii.es (N.L.-P.); jmasa@isciii.es (J.M.-C.); 3Laboratorio Regional de Salud Pública de la Comunidad de Madrid, 28055 Madrid, Spain; mpabeledo@salud.madrid.org; 4Servicio de Epidemiología, Consejería de Sanidad de la Comunidad de Madrid, 28009 Madrid, Spain; luis.garcia@salud.madrid.org; 5Centro Nacional de Epidemiología, Instituto de Salud Carlos III, 28029 Madrid, Spain; 6Departamento de Medicina Preventiva y Salud Pública, Universidad Autónoma de Madrid/IdiPAZ, 28049 Madrid, Spain

**Keywords:** laboratory surveillance, molecular epidemiology, genotyping, MF-NCR sequence, SH sequence, mumps virus, mumps

## Abstract

Mumps is a vaccine-preventable disease caused by the mumps virus (MuV). However, MuV has re-emerged in many countries with high vaccine coverage. The World Health Organization (WHO) recommends molecular surveillance based on sequencing of the small hydrophobic (SH) gene. Additionally, the combined use of SH and non-coding regions (NCR) has been described in different studies, proving to be a useful complement marker to discriminate general patterns of circulation at national and international levels. The aim of this work is to test local-level usefulness of the combination of SH and MF-NCR sequencing in tracing hidden transmission clusters and chains during the last epidemic wave (2015–2020) in Spain. A database with 903 cases from the Autonomous Community of Madrid was generated by the integration of microbiological and epidemiological data. Of these, 453 representative cases were genotyped. Eight different SH variants and thirty-four SH haplotypes were detected. Local MuV circulation showed the same temporal pattern previously described at a national level. Only two of the thirteen previously identified outbreaks were caused by more than one variant/haplotype. Geographical representation of SH variants allowed the identification of several previously undetected clusters, which were analysed phylogenetically by the combination of SH and MF-NCR, in a total of 90 cases. MF-NCR was not able to improve the discrimination of geographical clusters based on SH sequencing, showing limited resolution for outbreak investigations.

## 1. Introduction

Mumps is a vaccine-preventable disease caused by the mumps virus (MuV), which belongs to the *Orthorubulavirus* genus (*Paramyxoviridae* family). MuV virion is an enveloped particle containing a non-segmented negative single RNA strand, of 15,394 nucleotides (nt) in length. The MuV genome contains seven transcription units that encode the following: the nucleoprotein (N), the V/phospo-/I proteins (V/P/I), the matrix protein (M), the fusion protein (F), the small hydrophobic protein (SH), the haemagglutinin-neuraminidase protein (HN) and the RNA polymerase (L) [1]. Non-coding regions (NCR) of the MuV genome include the intergenic regions and the untranslated regions (UTRs) of each transcription unit.

MuV is transmitted by direct contact, through respiratory droplets and contaminated fomites. Swelling of the parotid glands and fever are the main clinical manifestations. Unspecific symptoms can occur beforehand, such as headache, malaise or anorexia. Other less frequent symptoms are orchitis, mastitis, oophoritis and pancreatitis. Encephalitis and aseptic meningitis can be unusual complications [2]. The infection is subclinical in up to 30% of cases in non-vaccinated individuals, especially in adults [3].

In Spain, the Measles, Mumps and Rubella vaccine (MMR vaccine) has been recommended for all children as part of the immunisation calendar, since 1981. Nowadays, the first dose is administered at 12 months of age, and a second dose between 3 and 4 years old [4]. As a result of this strategy, mumps incidence decreased drastically, from 211/100.000 inhabitants in 1982 to 35/100.000 inhabitants in 1992. Despite vaccine coverage exceeding 90% for both doses since 2003, MuV has caused cyclical epidemic waves (every 4–7 years): 2004–2009, 2010–2014 and 2015–2020, when the SARS-CoV-2 pandemic interrupted virus circulation due to the severe control measures adopted [5]. The reappearance of MuV, during recent years in countries with high vaccination coverage is a matter of concern, since it suggests a suboptimal effectiveness of current vaccines [2,6], which do not allow the elimination of viral circulation, although they provide a substantial reduction in complications [7]. Proposed explanations for this reduced efficacy of the vaccine include waning immunity [2,8] reduced cross-reactivity between genotypes [9,10] or between the circulating MuV and the vaccine strains, due to an antigenic drift from the time when the vaccine strains were isolated [11,12].

Sequencing of the SH gene is recommended by the World Health Organization (WHO) for MuV genotyping, as part of the epidemiologic surveillance necessary to control the disease [13]. According to this method, a total of 12 genotypes were recognized: A, B, C (including former genotype E), D, F, G, H, I, J, K (including former genotype M), L and N [14]. Other molecular tools based on the sequencing of NCRs have been proposed as additional markers, especially the region between the M and F genes (MF-NCR) [14,15,16]. Most recently, complete genome sequencing is becoming a fundamental tool for understanding infectious disease spread, identifying outbreak origins and tracing circulation patterns and transmission chains [16,17].

Genotype G has been the predominant genotype in Spain since it replaced genotype H, in 2005. A single SH variant (MuVi/Sheffield.GBR/1.05/) was detected in most cases notified from 2005 to 2015, so that NCR sequencing was necessary to reveal hidden circulation patterns [15]. However, during the last epidemic wave, genetic variability seemed to increase, and SH sequencing became useful again to trace MuV circulation at national [18] and international levels [19]. The combined use of SH and NCRs for the study of MuV circulation has been described at national and international levels in different studies [15,16,19,20]. However, data on the application of these tools to the study of MuV circulation at more reduced scales, or even to trace chains of transmission, are scarce [16].

The aim of this paper is to test the usefulness of the combination of SH and MF-NCR sequencing at a local level (Autonomous Community of Madrid) to trace hidden transmission clusters and chains, in the context of mumps surveillance during the last epidemic wave (2015–2020) in Spain, by integrating microbiological and epidemiological data.

## 2. Materials and Methods

### 2.1. Sample Database and Period of Study

A database was generated integrating the microbiological and epidemiological data of 903 mumps cases confirmed by RT-PCR in clinical samples tested at the Laboratorio Regional de Salud Pública (LRSP) of the Autonomous Community of Madrid (CM), and subsequently confirmed at the Centro Nacional de Microbiología (CNM) of the Instituto de Salud Carlos III (ISCIII), during the period of study (2015–2020). The database included the following variables provided by the LRSP-CM, CNM-ISCIII and Dirección General de Salud Pública de la CM: sex, date of birth, sample collection date, sample type, date of onset, sample identification number, outbreak identification (ID), sanitary-area code, district, spatial coordinates (military technical units, UTM) and vaccination status. One sanitary area comprises several districts.

Of 903 confirmed cases, 128 were associated with an outbreak ID. The selection criterium for genotyping was at least one case per week and per sanitary area code. When a selected case could not be genotyped, it was replaced by another of the same week and area.

### 2.2. Nucleic Acid Extraction, Genetic Amplification and Sequencing

Nucleic acid extraction was, carried out with an automatic extractor (QIAsimphony, QIAGEN, Hilden, Germany), using a commercial kit (QIAsymphony DSP Virus/Pathogen Midi Kit (96 preps); QIAGEN, Hilden, Germany).

Amplification of the SH gene (316 nt) was carried out according to a previously described protocol [21]. The MF-NCR fragment (452 nt) was additionally sequenced in a subset of samples, as previously published [15].

PCR products were purified with Illustra ExoProStar 1-Step (GE Health Care Life Science, Freiburg, Germany) and sequenced with the ABI Big Dye Terminator Cycle Sequencing Kit (Applied Biosystems, Branchburg, NJ, USA), using the corresponding forward and reverse primers.

### 2.3. Genetic and Phylogenetic Analysis, Variant Classification and Geographical Representation

Sequences were edited using BioEdit v.7.2.5 [22], and aligned with MAFFT v.7 [23]. Haplotypes (defined as a set of identical sequences) were identified using BioEdit v.7.2.5 [22]. Nucleotide differences were identified using MuVi/Sheffield.GBR/1.05/as reference (ON148331). Those haplotypes detected for six months or more and/or spreading to three or more Spanish provinces or different countries were considered variants, as described in previous works [19]. Derived SH haplotypes are those that shared a characteristic variant substitution and an additional change. Sequence fragments of the different genes were concatenated using BioEdit v.7.2.5 [22]. Phylogenetic analysis was performed using the maximum likelihood method (ML) with IQ-TREE software, via the webserver (W-IQ-TREE) [24]. Branch support was calculated using the ultrafast bootstrap approach (UFboot) [25]. Phylogenetic trees were edited using Figtree v.1.4.4. [26]. Geographical analysis and representation was carried out by QGIS desktop v.3.30.0 [27], at district level.

### 2.4. GenBank Accession Number

The SH and MF-NCR sequences used in this study have been deposited in GenBank (Appendix A).

### 2.5. Ethics Statement

All samples were collected through the Mumps Microbiological Surveillance Programme of the CNM-ISCIII, according to the requirements of the Spanish biomedical research law (Ley 14/2007 de Investigación biomédica). The protocol was approved by the Comité de Ética de la Investigación del Instituto de Salud Carlos III (approval. reference code: CEI PI 35-2015).

## 3. Results

Of the total confirmed cases, 55.5% were male and 42.2% female, while the remaining 2.3% did not have available data. The mean age was 24.2 years (range 1–72 years). The most common sample was saliva (*n* = 895), followed by nasopharyngeal swab (*n* = 8). Vaccination status was known for 458 cases (50.7%). Of those, 34.3% had received one dose, 51.2% two doses, and 5.7% three doses. The mean number of days between symptom onset and sample collection was 3 days (range 0–17).

A total of 453 MuV positive cases were selected for genotyping. They represent 7.5% of the 6,081 declared in the CM during the period of study. The representativeness of the sample increased from 3.7% in 2015 to 21.6% in 2019. The COVID-19 pandemic caused a total disruption of the virus circulation in early 2020 (Figure 1). Seventy-three of them were associated with fifteen different outbreak ID codes. Eight different SH variants were identified, all of them previously described [18,19]. In addition, 34 different SH haplotypes were detected during the period of study. Of these, MuVs/Madrid.ESP/10.18/3 is des-scribed for the first time.

### 3.1. Temporal Distribution of MuV SH Variants

The analysis of SH sequences showed how the MuVi/Sheffield.GBR/1.05/-variant was predominant until the beginning of 2016, when the MuVs/Avila.ESP/11.16/-variant appeared and replaced it. At the end of 2016, two other SH variants emerged: MuVs/NewYork.USA/45.15/, which co-circulated with MuVs/Avila.ESP/11.16/ during 2017, and MuVs/Madrid.ESP/50.16/2, which became dominant during 2018. One case of this variant was detected in week 42, eight months before the case that gave the name to the variant in previous publications [15,18]. We decided to keep the name the same as in previous works, to avoid confusion. MuVi/Sheffield.GBR/1.05/ re-emerged mid-2017, co-circulating with MuVs/Avila.ESP/51.18/, MuVs/New_Jersey.USA/20.10/, MuVs/Salamanca.ESP/24.19/ and MuVs/Tarragona.ESP/20.11/, until the onset of the COVID-19 pandemic (Figure 1). This succession of variants was concordant with that previously described at a national level [18].

MuVs/Madrid.ESP/50.16/2 and MuVs/Tarragona.ESP/20.11/ were detected for the first time in the CM during the period of study, as well as MuVi/Sheffield.GBR/1.05/, when it remerged in mid-2017. The remaining six were found in different Spanish provinces before that in the CM [18]. MuVs/Avila.ESP/11.16/, MuVs/New_Jersey.USA/20.10/, MuVs/Tarragona.ESP/20.11/ and MuVs/Salamanca.ESP/24.19/ were introduced in Madrid a few weeks after they were detected for the first time in other locations, while MuVs/Avila.ESP/51.18/ was not detected until ten weeks later [18].

### 3.2. Local Circulation, Geographical Representation and Phylogenetic Analysis

The SH variants and haplotypes did not show a specific pattern of aggregation in circumstances of co-circulation of different variants (Figure 2), apart from the previously described temporal pattern of succession (Figure 1). Geographical representation allowed the identification of some exceptions, as clusters (groups of contemporary cases of a discordant variant or haplotype identified in the same district (Figure 2)).

A cluster formed by four cases of the MuVs/NewYork.USA/45.15/ SH variant was observed in 2016 in the district of Fuenlabrada, in the south of the Community of Madrid (Figure 2, year 2016). Additionally, another SH haplotype, MuVs/Madrid.ESP/13.17/4, was identified within this cluster, sharing with MuVs/NewYork.USA/45.15/ the characteristic nt substitution C248T, but also G179T, which suggests a possible local-evolution origin, according to the evolution rate (1.7 × 10^−3^ substitution/site/year) [28].

Another cluster of seven cases of a new SH haplotype (MuVs/Madrid.ESP/10.18/3) was found circulating in the south-eastern area of the Community of Madrid, over 19 weeks (Figure 2, year 2018). This haplotype shows the nt substitution (G122A), in addition to the characteristic two differences of the MuVs/Madrid.ESP/50.16/2 SH variant (A226T and T247C), suggesting an emergence by local evolution (Figure 3, Panel B) according to the evolution rate. Although these cases were not initially linked with an outbreak ID, subsequent investigations revealed that they all had a family link.

Another cluster of seven cases of another SH haplotype, MuVs/Madrid.ESP/19.19/3, was detected in the Alcalá de Henares district, in 2019 (Figure 2, year 2019). It shows the nt substitution C13A with the MuVs/Avila.ESP/51.18/ SH variant, and also exhibits A284C, suggesting again an origin by local evolution, according to the evolution rate. As in the previous clusters, an epidemiological link among these cases has not been found.

Finally, another cluster of six cases belonging to the MuVs/New_Jersey.USA/20.10/ SH variant, not previously linked, was observed in 2019 in the district of Torrejón de Ardoz (Figure 2, year 2019). The cases were between 14 and 16 years old, and clustered around a high school. This SH variant caused a massive outbreak in the city of Segovia (90 Km from Madrid) in mid-2019, and then spread to Madrid [18].

A total of 90 MF-NCR sequences from the previously described clustered cases were obtained, in order to explore correlations with geographical clustering (Appendix A). Subsequently, the potential usefulness of this marker for establishing relationships among cases of mumps at the local level was studied. The phylogenetic analysis of these MF-NCR sequences (Figure 3, Panel A) showed a smaller number of clades than the SH sequences of the corresponding cases (Figure 3, Panel B), because the MF NCR has less nt variability than SH, as previously cited [2]. Nevertheless, the tree topology of MF-NCR and SH concatenated sequences showed different clusters associated with the MuVi/Sheffield.GBR/1.05/, MuVs/Avila.ESP/11.16/ and MuVs/Madrid.ESP/50.16/2 SH variants, during the period of study (2015–2020) (Figure 3, Panel C).

An identical MF-NCR sequence was shared by the MuVs/Madrid.ESP/50.16/2 SH variant and the MuVs/Madrid.ESP/10.18/3 SH haplotype, as well as the MuVs/Avila.ESP/51.18/ SH variant and MuVs/Madrid.ESP/19.19/3 SH haplotype, which is compatible with the hypothesis of local evolution as the origin of derived SH haplotypes (Figure 3, Panel A). Regarding MF-NCR sequences associated with the MuVs/New_Jersey.USA/20.10/ SH variant, of the cluster from the Torrejón de Ardoz district, as well as those associated with the MuVs/NewYork.USA/45.15/ SH variant of the cluster of the Fuenlabrada district, these variants were identical to those identified in the rest of the CM and, consequently, a correlation with geographical clustering was not established. Moreover, all MF-NCR sequences associated with the MuVs/NewYork.USA/45.15/ SH variant were grouped in an independent and unique phylogenetic clade with the SH haplotype MuVs/Madrid.ESP/13.17/4, which shares the characteristic nt substitution, C248T, of MuVs/NewYork.USA/45.15/ plus an additional variation, suggesting an evolutionary origin (Figure 3, Panel A).

### 3.3. Molecular Analysis of Previously Identified Outbreaks

A total of 15 different clusters of cases previously grouped by epidemiological criteria, with an outbreak identification, were considered. Eleven of them were small outbreaks of fewer than four cases and up to four weeks of duration. The remaining four (2016/418, 2017/252, 2019/640 and 2019/677) were significantly bigger, with seven to fourteen cases, and extended up to fifteen weeks (2016/418). Educational centres were the most common among those with information. The same SH variant was identified in all the cases linked to the same outbreak ID, in 13 outbreaks (Table 1).

All SH variants coincided in temporal and geographical distribution with the general circulation pattern of MuV described above for the CM. The most common SH variant was MuVs/Avila.ESP/11.16/ (seven outbreaks), followed by MuVi/Shffield.GBR/1.05/(five outbreaks). Only one outbreak (2017/252) was caused by an SH haplotype (MuVs/Madrid.ESP/13.17/4). In the remaining two, 2016/443 and 2017/49, the MuVs/Avila.ESP/11.16/ SH variant was found, together with an additional different haplotype that showed the characteristic A226T substitution of the MuVs/Avila.ESP/11.16/ SH variant, together with an additional change (Table 2), suggesting an origin by local evolution in both cases. The finding of the same MF-NCR sequence, both on the MuVs/Avila.ESP/11.16/ SH variant and the accompanying derived haplotypes, supports this hypothesis (Appendix A).

## 4. Discussion

Previous works of the group focused on the use of NCRs to complement SH sequencing, to trace MuV circulation patterns in Spain and Europe [18,19]. The present study complements these works, by focusing on the use of these markers for case clustering at a local level. Most variants of MuV detected in the Autonomous Community of Madrid found in this work are concordant with those previously described for the whole territory of Spain, after genotype G emerged in 2005 [15,18,28]. MuVi/Sheffield.GBR/1.05/ was the dominant variant before the period of the study, as in the rest of Spain, until it was detected for the last time in Spain in early 2016, in Madrid. This variant re-emerged in Madrid in week fifteen of 2017, being detected for the first time outside Madrid, in Murcia, in week 41. The MuVs/Avila.ESP/11.16/-variant emerged in Ávila, to be detected in Madrid only two weeks after, which occurred again in 2018 with MuVs/Avila.ESP/51.18/. The MuVs/Madrid.ESP/50.16/2-variant emerged in Madrid in 2016, to be detected in Valencia ten weeks after, before spreading to the rest of the country. The MuVs/NewYork.USA/45.15/-variant was detected for the first time in Ciudad Real, in week 16, 2016, entering Madrid in week 33. MuVs/Tarragona.ESP/20.11/ was detected for the first time in Madrid in late 2018, and was detected in Navarra more than thirty weeks after. MuVs/Salamanca.ESP/24.19/ emerged in week 24, 2019, in Salamanca, then causing an outbreak in Galicia, starting in week 34, and reaching Madrid in week 39. Finally, the MuVs/New_Jersey.USA/20.10/-variant emerged in Segovia, causing a big outbreak starting in week 34, and was then detected in Madrid for the first time, in week 38. The Autonomous Community of Madrid is a small but highly populated region, with an intensive population exchange with the rest of the country, and where an overlap of internal circulation and imported events drives MuV circulation. This justifies the choice of this community to study these molecular tools at a local level, in the same framework previously described for the whole territory of Spain.

Despite Madrid having an intense international passenger traffic, only two of the eight detected variants seemed to enter Spain via Madrid. The other six were detected for the first time in other locations with many fewer intense international connections. However, some of them, such as Segovia, Salamanca or Avila are historical cities, which receive thousands of tourists every year. Interestingly, all of the variants finally extended to Madrid.

The cases included in the study represented the 7.5% of all those declared in the territory during the period of study, which is a good representation, considering the pressure suffered by the laboratory due to the high MuV circulation observed during these years in Spain. Representativeness could have even been improved, however, the severe public health measures implemented to control the COVID-19 pandemic had a disruptive impact, both on the capacity of the laboratory and on viral circulation, which seemed to be interrupted. Most confirmed cases with data on vaccination status corresponded to fully vaccinated people (two or more administrated doses), according to the general trends described for Spain [5].

SH and MF-NCR molecular tools supported the small outbreaks detected by epidemio-logical investigation, since all cases shared the same sequences for both genomic regions, in most outbreaks. Interestingly, the onset of single mutations on the SH coding sequence were observed in the course of two outbreaks (2016/443 and 2017/49), although they were quite limited in duration and extension (two and three weeks, respectively). However, nucleotide differences on the MF-NCR sequence were not observed in any outbreak, which coincides with its lower variability [2]. Therefore, the interpretation of single mutations in cases regarding the linking and tracing of chains of transmission of MuV studies should be treated with caution.

The geographical representation of SH variants and haplotypes revealed clusters of cases which had not been previously identified in the epidemiological studies at the moment when they occurred. Only one compatible epidemiological link was found subsequently, in two outbreaks, the first one in a family, and a second one associated with an educational centre. However, MF-NCR did not differentiate this geographical clustering, and appears to lack the resolution to complement SH for tracing chains of transmission and cluster discrimination of linked cases at a local level. In this study, MF-NCR only contributed to tracing local evolution after the onset of point mutations on the SH gene.

The study of viral variants is a fundamental tool for the surveillance of infectious diseases, including those vaccine-preventable, such as mumps. With this purpose, WHO recommends standardized procedures for MuV genotyping, based on SH gene sequencing [13]. However, this molecular marker turned out to be insufficient for situations of dominance of single genotypes or variants [15]. For this purpose, the use of additional genomic regions, such as NCRs, to complement SH, in order to improve the resolution, has been proposed in the last years [15]. The use of NCRs has proved to be useful for establishing MuV circulation patterns in big territories [18,19]. However, data on the use of NCRs for establishing circulation patterns, characterizing outbreaks, and tracing chains of transmission at a local scale are scarce. The results shown in this work are concordant with previous studies showing MF-NCR as a useful marker to complement SH sequencing for the study of the general patterns of MuV circulation in Spain [18] and Europe [19]. However, the use of MF-NCR does not seem to provide a significant improvement in SH sequencing for case clustering at a local level, which requires the study of the whole genome, as has been previously described [17,29]. Although next-generation sequencing is still not widely available, the inclusion of these techniques in the regular protocols for molecular surveillance of different viral diseases, such as influenza or COVID-19 [29], is accelerating their implementation in many laboratories.

## Figures and Tables

**Figure 1 viruses-15-02420-f001:**
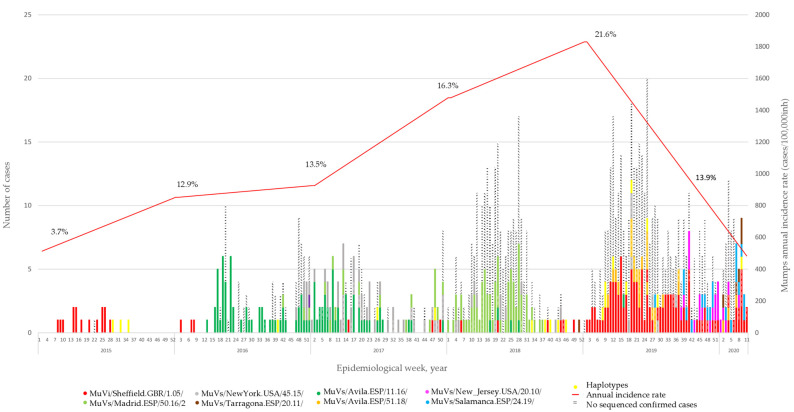
Temporal distribution of mumps virus SH variants and haplotypes in the Community of Madrid from 2015 to 2020. The *Y*-axis shows the number of confirmed cases (PCR-positive) and the annual incidence rate of mumps per 100.000 inhabitants in the CM. The *X*-axis represents the epidemiological weeks, grouped into years. The percentages represent positive cases detected in the CNM-ISCIII out of the total number of cases reported in the same year. Cases belong to the variants and haplotypes are coloured according to the legend assignation.

**Figure 2 viruses-15-02420-f002:**
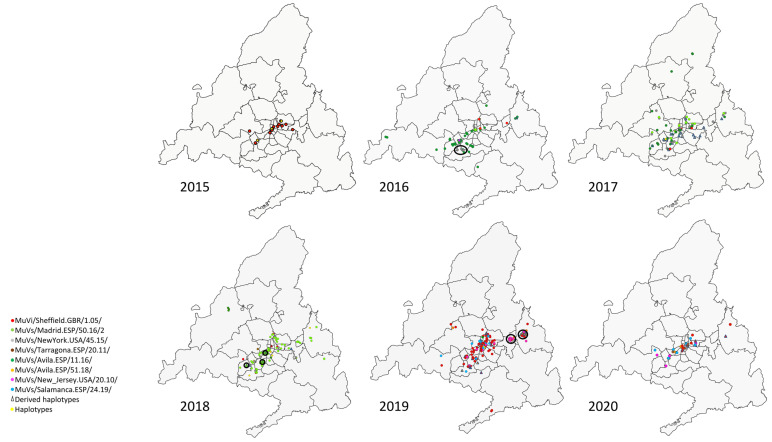
Geographical distribution of mumps virus SH variants and haplotypes from 2015 to 2020 in the Community of Madrid. Each case is represented by a coloured spot, according to the associated SH variant, as indicated in the legend. Derived haplotypes of SH variants are represented by triangles following the same coloured code. Other haplotypes are represented by a yellow spot. Clusters are marked with a black circle.

**Figure 3 viruses-15-02420-f003:**
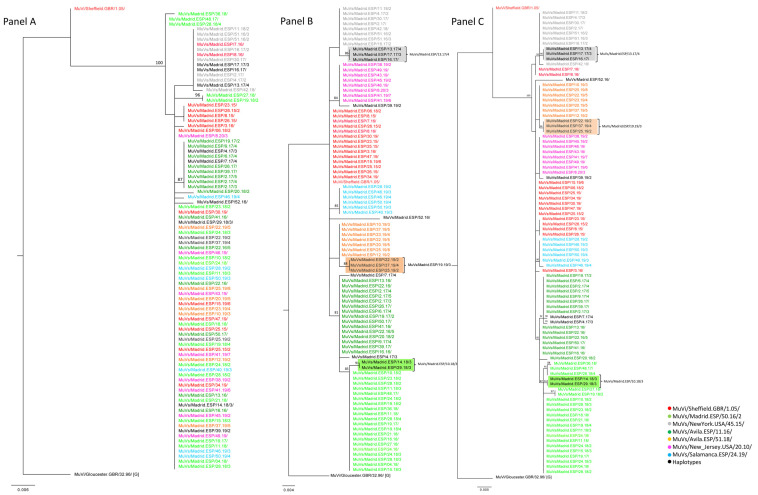
Phylogenetic analysis of the sequences obtained from cases geographically clustered. (**A**) Phylogenetic tree of MF-NCR sequences (452 nt). (**B**) Phylogenetic tree of SH sequences (316 nt). (**C**) Phylogenetic tree of MF-NCR-SH concatenated sequences (768 nt). Phylogenetic trees were made using the maximum likelihood method in W-IQ-TREE, using GTR + I, TN93 and GTR as substitution models, respectively. MuVi/Sheffield.GBR/1.05/ (ON148331) and MuVi/Gloucester/32.96/[G] (AF280799) were used as outgroup. Each name sequence is coloured according to the SH variant assignation.

**Table 1 viruses-15-02420-t001:** SH variants and haplotypes of mumps cases belonging to previously described outbreaks.

Outbreak ID (Year/Outbreak)	No. of Cases	No. of Genotyped Cases	Epidemiological Outbreak Data	Variant/Haplotype	Percentage of Cases Which Belong to the Variant/Haplotype
2016/321	2	2	Educational centre	MuVi/Sheffield.GRB/1.05/	100%
2016/356	2	2	-	MuVs/Avila.ESP/11.16/	100%
2016/353	3	2	-	MuVs/Avila.ESP/11.16/	100%
2016/411	2	2	-	MuVs/Avila.ESP/11.16/	100%
2016/418	14	3	Educational centre	MuVs/Avila.ESP/11.16/	100%
2016/441	2	2	-	MuVs/NewYork.USA/45.15/	100%
2016/443	4	4	Educational centre	MuVs/Avila.ESP/11.16/	75%
				MuVs/Madrid.ESP/4.17/3	25%
2017/49	3	3	Educational centre	MuVs/Avila.ESP/11.16/	66.7%
				MuVs/Madrid.ESP/7.17/4	33.3%
2017/169	2	2	-	MuVs/Avila.ESP/11.16/	100%
2017/252	7	4	Sociocultural association	MuVs/Madrid.ESP/13.17/4	100%
2018/424	4	3	Educational centre	MuVs/Madrid.ESP/50.16/2	100%
2018/446	3	2	-	MuVs/Madrid.ESP/50.16/2	100%
2019/640	7	2	Educational centre	MuVi/Sheffield.GRB/1.05/	100%
2019/677	11	2	Educational centre	MuVi/Sheffield.GRB/1.05/	100%
2019/679	3	2	Educational centre	MuVi/Sheffield.GRB/1.05/	100%

**Table 2 viruses-15-02420-t002:** SH nucleotide differences of mumps cases belonging to 2016/443 and 2017/49 outbreaks. MuVi/Sheffield.GBR/1.05/ SH sequence was used as reference (EU597478).

Outbreak ID	SH Variant/Haplotype	*n*	SH Differences (nt)		
2016/443	MuVs/Avila.EPS/11.16/	3	-	A226T	-
	MuVs/Madrid.ESP/4.17/3	1	G56A	A226T	-
2017/49	MuVs/Avila.EPS/11.16/	2	-	A226T	-
	MuVs/Madrid.ESP/7.17/4	1	-	A226T	G261T

## Data Availability

Data are contained within the article.

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
