# Peer review of "Investigating Local Patterns of Mumps Virus Circulation, Using a Combination of Molecular Tools"

_viruses, 2023, doi:10.3390/v15122420_

Round 1
Reviewer 1 Report
Comments and Suggestions for Authors
Comments on the Quality of English LanguageThe quality of the writing is clear with minor editing required.
Author Response
Answers to referee 1
- Given that the conclusion of the study was that the MF-NCR sequence did improve the molecular resolution of similar mumps variants, I suggest that the title should be changed. Perhaps “Investigating” rather than “Revealing….with a combination of molecular tools.”
We agree with the change.
- Likewise, the final sentence of the abstract is a conclusion that is based on the literature, not the current study. I think it should be moved into the introduction of the abstract, specifically line 27 immediately before the sentence starting with “The aim of this work…”.
The final sentences has been moved to line 27-28.
- The visual resolution of all figures needs to be improved. The text was illegibly small but was not made more legible by zooming in the figure. This includes the legends of all figures (variants) and the axes of Figure 1. In Figure 2, the triangles (derived haplotypes) could not be distinguished from the circles. In Figure 3, the text of the sequence names in the phylogenetic trees could not be clearly read. In all cases, this made it difficult to visually confirm the statements the authors made in the text regarding the findings.
All figures have the maximum resolution in word format. However, when manuscript is transformed into PDF format quality get worse. In order to solve that, each figure was individually attached in the manuscript submission.
- In section 2.1, the authors describe the data within their database of RT-PCR confirmed mumps cases which included many demographic variables, none of which were used in this study, with the exception of geographic information. The description of the variables that are not used here should be removed as they are not within the stated aim of the current study. (As an aside, I encourage the authors to make use of this epi-lab linked data to conduct an in depth study of possible risk factors. In particular, linking the vaccination status to genotype or case status would be very interesting.
A paragraph about the epidemiological data of the cases included in the study, according to the variables described in material and methods has been included in the results section (lines 149-154) and in the discussion section (lines 372-374).
- Figure 1: Is the incidence provided the national incidence or only for CM? I suspect this is the national incidence. Please clarify in the figure caption.
The given annual incidence rate of mumps corresponding to Community of Madrid. In order to avoid a misunderstanding, it has been detailed in line 178.
- Lines 174 – 180: it seems to me that the study authors are placing their findings within the broader national context which relies on data not presented in this study. Appropriate citations should be placed, and perhaps more appropriately, this should be moved to the Discussion.
We agree with the review´s suggestion. Citations have been included in the current line 184 and line 187. In addition, at the beginning of the discussion, a new paragraph detailing data of Madrid in the context of Spain have been included (lines 342-356, 365).
- In the geographic analysis (Figure 2), in addition to classifying SH gene sequences by variants and haplotypes, the authors use “derived haplotypes of SH variants.” Please provide the definition.
Derived SH haplotype definition has been included in lines 129-130.
- Please identify in the manuscript, and particularly in the caption for Figure 3, how long the two sequencing regions are (SH gene and MF-NCR).
The length of both fragments has been detailed in lines 115-116 in 2.2 section and in the Fig. 3 caption (lines 264-265).

Reviewer 2 Report
Comments and Suggestions for Authors
Gavilan and colleagues here report on an attempt to utilize MF-NCR sequencing to complement SH sequencing in an effort to better map mumps transmission chains at a localized level using recent mumps outbreaks in Spain. The authors examine a representative sampling of outbreak samples using both approaches and found that MF-NCR sequencing does not improve the resolution for defining chains of transmission in outbreaks – which may be expected, based on some of the authors statements about the variability of the MF-NCR region. Overall, the paper is well written and should be considered for publication once the authors address some minor points below:
Line 56-57: This should be noted that 30% subclinical infection in unvaccinated persons. The rate of subclinical infection in vaccinated persons has not yet been definitively ascertained.
Line 60: Change “inoculated” to “administered”
Line 62: Change “211/100.00” to “211/100.000”
Line 70-71: It’s unclear to me how “reduced genotype cross-reactivity” and “antigenic drift” are not the same phenomenon. Wouldn’t antigenic drift lead to reduced genotype cross-reactivity? Suggest combining as one topic unless additional information is provided.
Fig 1: Increase figure resolution.
Fig 2: Increase figure resolution and size. It’s extremely difficult to discern spots from triangles on the maps.
Line 198-199: What is the likelihood of co-evolution compared to co-circulation? Possible this case was imported from a different area separately from the other cases.
Fig 3: Improve figure resolution.
Lines 277-278: Should “away” be “amid” or “between”? Unclear in this context.
Line 280-281: If MF-NCR has less nt variability and this is known, then sequencing this region would give you less resolution of the genetic sequences and might run the risk of misclassifying cases as linked when they are in fact not.
Section 3.2: Several times throughout the authors note that mutations in the SH gene are indicated to occur by local evolution. Does this make sense with the rate of mutations in RNA virus transmission given the number of cases? I think this should be mentioned in the text as supporting evidence if so.
Line 383-384: Agree here with the authors point that whole genome sequencing might be the best alternative to high resolution tracing of transmission chains for mumps. This has been very successful for measles chains of transmission in elimination settings.
Comments on the Quality of English Language
Well written paper. A few changes to word usage suggested.
Line 60: Change “inoculated” to “administered”
Lines 277-278: Should “away” be “amid” or “between”? Unclear in this context.
Author Response
Answers to referee 2
Line 56-57: This should be noted that 30% subclinical infection in unvaccinated persons. The rate of subclinical infection in vaccinated persons has not yet been definitively ascertained.
Following the review comment, the statement “in non-vaccinated individuals” has been included in line 57.
Line 60: Change “inoculated” to “administered”
The term “Inoculated” has been changed by “administered” and the term “is administered” has been removed from the same line to avoid word repetition.
Line 62: Change “211/100.00” to “211/100.000”.
An additional zero was added.
Line 70-71: It’s unclear to me how “reduced genotype cross-reactivity” and “antigenic drift” are not the same phenomenon. Wouldn’t antigenic drift lead to reduced genotype cross-reactivity? Suggest combining as one topic unless additional information is provided.
In reference 11 authors find that vaccine-induced antibodies neutralise MuV strains close to the time when vaccine strains were isolated. However, neutralisation of modern isolates is not optimal, irrespective of the genotype, suggesting antigenic drift as a cause. Consequently, reduced genotype cross-reactivity and antigenic drift are different concepts in this context and we prefer to consider them independently. We have detailed the sentence in lines 70-72 to make it clearer.
Fig 1: Increase figure resolution. Fig 2: Increase figure resolution and size. It’s extremely difficult to discern spots from triangles on the maps. Fig 3: Improve figure resolution.
All figures have the maximum resolution in word format. However, when manuscript is transformed into PDF format quality get worse. To solve that, each figure was individually attached in the manuscript submission.
Line 198-199: What is the likelihood of co-evolution compared to co-circulation? Possible this case was imported from a different area separately from the other cases.
See comment below in section 3.2 on local evolution according to the MuV SH evolution rate.
Lines 277-278: Should “away” be “amid” or “between”? Unclear in this context.
In order to clarify the sentence, “away” has been replaced by “among” in current line 285.
Line 280-281: If MF-NCR has less nt variability and this is known, then sequencing this region would give you less resolution of the genetic sequences and might run the risk of misclassifying cases as linked when they are in fact not.
We agree with the reviewer comment. For this reason, MF-NCR sequences are not used alone, but in combination with SH. Reviewer is right to find that sequences could be linked erroneously even using SH-MF-NCR concatenated sequences. For this reason, we have tried to be very strict with the tone of the conclusions when we analyse this data, and we conclude that maximum resolution can be only achieved by the use of complete genomes.
Section 3.2: Several times throughout the authors note that mutations in the SH gene are indicated to occur by local evolution. Does this make sense with the rate of mutations in RNA virus transmission given the number of cases? I think this should be mentioned in the text as supporting evidence if so.
Rate of mutation for SH gene is 1.71x10-3 substitutions/site/year, which is 0,54036 substitutions/year for the whole SH gene, per single case. As an example, for the calculation for MuVs/Madrid.ESP/10.18/3, as there are 7 cases we should expect 3,78252 substitutions/year for the cluster. As the cluster lasted 19 weeks, we should expect 1,38 changes, which is in the range of the real value of 1.
Any number of changes under the theoretical is compatible with local evolution if we consider that there are additional cases that has not been recorded. Some statements related to this have been included in the text: “according to the evolution rate” (1,7 10-3 substitution/site/year) in current lines 205, 210 and 273.
Line 383-384: Agree here with the authors point that whole genome sequencing might be the best alternative to high resolution tracing of transmission chains for mumps. This has been very successful for measles chains of transmission in elimination settings.
We agree and thanks reviewer´s comment.
